# Impact of the COVID-19 Pandemic on Antibiotic Prescribing by Dentists in Galicia, Spain: A Quasi-Experimental Approach

**DOI:** 10.3390/antibiotics11081018

**Published:** 2022-07-29

**Authors:** Almudena Rodríguez-Fernández, Olalla Vázquez-Cancela, María Piñeiro-Lamas, Adolfo Figueiras, Maruxa Zapata-Cachafeiro

**Affiliations:** 1Department of Preventive Medicine and Public Health, University of Santiago de Compostela, 15786 Santiago de Compostela, Spain; almudena.rodiguez@usc.es (A.R.-F.); maria.pineiro@usc.es (M.P.-L.); adolfo.figueiras@usc.es (A.F.); maruxa.zapata@usc.es (M.Z.-C.); 2Department of Preventive Medicine, Santiago de Compostela University Teaching Hospital, 15706 Santiago de Compostela, Spain; 3Consortium for Biomedical Research in Epidemiology & Public Health (CIBER en Epidemiología y Salud Pública-CIBERESP), University of Santiago de Compostela, 15786 Santiago de Compostela, Spain

**Keywords:** antibiotics, COVID-19, dentists, prescriptions, AWaRe classification, primary care

## Abstract

Background: Antibiotic resistance is one of the most pressing public health problems. Health authorities, patients, and health professionals, including dentists, are all involved in its development. COVID-19 pandemic restrictions on dental care may have had repercussions on antibiotic prescribing by dentists. The aim of this study was to assess the impact of the COVID-19 pandemic on antibiotic prescribing by dentists, and to review antibiotic consumption according to the WHO Access, Watch, Reserve classification. We conducted a natural, before-and-after, quasi-experimental study, using antibiotic prescription data covering the period from January 2017 to May 2021. A segmented regression analysis with interrupted time series data was used to analyse the differences between the numbers of defined daily doses (DDD) of antibiotics prescribed monthly. The outcomes showed an immediate significant decrease in overall antibiotic prescribing by primary-care dentists during lockdown, followed by a non-significant upward trend for the next year. This same pattern was, likewise, observed for Access and Watch antibiotics. COVID-19 pandemic restrictions on dental care influenced the prescription of antibiotics. During confinement, an initial decrease was observed, this trend changed when in person consultations were recovered. It might be beneficial to analyse the prescription of antibiotics using the WHO AWaRe classification, in order to monitor their appropriate use.

## 1. Introduction

Antibiotic resistance is one of the most pressing public health problems worldwide due to its impact on mortality, morbidity, and healthcare costs [1]. The excessive and inappropriate use of antibiotics contributes a great deal to this problem. Health authorities, patients, and health professionals are all involved in this misuse [2]. Dentists prescribe 10% of all antibiotics consumed [3,4] and it is estimated that, from this, only 30% are correctly prescribed [5,6], since in dentistry most of the processes can be resolved by local treatments, and antibiotics would only be indicated on limited occasions [7].

The COVID-19 pandemic has had repercussions on dental care, due to the risk of cross-contamination in dental clinics due to the characteristics of dental procedures, during which aerosols are generated, and due to the close proximity of the professional to the patient [8]. During the first months of the pandemic in Spain, medical care was dispensed by means of telephone consultations in which the professional decided whether a physical medical visit was necessary [9]. In patients with symptoms compatible with COVID-19 or suspicion of infection, and with pain or inflammation of dental origin, the visit had to be postponed for a minimum of two weeks, during which time dental professionals had to resort to pharmacological treatment with antibiotics and/or analgesics [10]. This telephone-based care, coupled with the initial difficulty of obtaining personal protective equipment, and the fear felt by professionals and patients alike of contracting the disease during treatment, may well have served to increase the number of antibiotic prescriptions issued without an accurate diagnosis of infection, since it is extremely complicated to make a differential diagnosis based exclusively on information provided by the patient [9,11,12,13].

Currently, the impact of the COVID-19 pandemic on the short-term prescribing of antibiotics is unknown. Hence, the principal aim of this study was to assess the impact of the pandemic on antibiotic prescribing by dentists whose practices come within the Spanish National Health System (NHS) in north-west Spain; the secondary aim was to try to assess the impact of COVID-19 on the prescribing of those antibiotic groups that are most closely related with the development of antibiotic resistance.

## 2. Results

### 2.1. Antibiotic Prescribing by the WHO AWaRe Classification

Primary-care dentists prescribed a mean of 35344 DDD of antibiotics before confinement, approximately 6% of which belonged to the Watch group, as shown in Table 1. No antibiotic was prescribed in the Reserve group. The mean DDD values of Access and Watch group antibiotics were similar before, during, and after confinement. Amoxicillin, amoxicillin combined with clavulanic acid, azithromycin, and clindamycin were the antibiotics most used by dentists. Azithromycin, belonging to the Watch group, was the third most prescribed antibiotic (Appendix A).

### 2.2. Impact of the COVID-19 Pandemic on Antibiotic Prescribing

The monthly antibiotic prescribing trend from January 2017 to May 2021 for primary-care dentists is shown in Figure 1 and Table 2. As can be seen, after seasonal adjustment, there were changes in the antibiotic prescribing trend prior to the outbreak of the pandemic (B = −42.39). On the emergence of COVID-19, a significant decrease in antibiotic prescribing (B = −3572.73; *p* < 0.005) was observed during lockdown (equivalent to a decrease of 8.92% (95% CI: −3.80%; 14.13%)), followed by an immediate non-significant decrease (B = −1363.82) with respect to the pre-pandemic period, and a subsequent upward trend in the long term (B = 142.74).

### 2.3. Impact of the COVID-19 Pandemic on the WHO Access Group 

Figure 2 depicts the monthly prescribing trend in DDD among primary-care dentists for the group of antibiotics listed as Access by the WHO. Table 2 shows a statistically significant reduction (B = −2976.50, *p* = 0.0133) in antibiotic prescribing during lockdown (8.00% (95% CI: 2.68%; 13.46%) decrease) followed by an immediate non-significant reduction in prescribing (B = −991.28) and a non-significant upward trend in the long term (B = 136.05).

### 2.4. Impact of the COVID-19 Pandemic on the WHO Watch Group 

Figure 3 depicts the monthly prescribing trend in DDD of Watch antibiotics. During lockdown, a statistically significant reduction (B = −651.02; *p* < 0.0001) in antibiotic prescribing was observed (decrease of 20.16% (95% CI: 14.05%; 26.26%)). Whereas the emergence of COVID-19 had a significant impact (B = −688.91) on the trend, with a reduction in prescribing (*p* = 0.0002); in the long-term, a non-significant increase was evident (B = 29.15). 

## 3. Discussion

The data yielded by this quasi-experimental study indicate that the COVID-19 pandemic had an important impact on antibiotic prescribing by NHS dentists in north-west Spain. In March 2020, coinciding with the Spanish Government’s declaration of a state of alarm and confinement, a significant decrease was observed in monthly antibiotic prescriptions, which was maintained during lockdown and followed by an upward trend across the following year. This pattern was, likewise, generally observed for antibiotics and for those in the Access and Watch groups, with 8.92%, 8.00%, and 20.16% respectively.

The initial decrease observed may be a consequence of the measures adopted during the first phase of confinement, in which care was dispensed by telephone consultation and, in rare cases, in person [9]. Hence, the suspension of non-urgent surgical treatments (such as extractions of asymptomatic teeth) may have led to a decrease in the prescribing of antibiotics associated with these types of procedures, and of Watch-type antibiotics in particular [15,16,17]. Patients’ fear of visiting dental clinics due to the risk of becoming infected with COVID-19, the smaller number of emergencies, and the increase in self-medication could also account for this initial decrease [18,19,20]. 

The sharp drop observed during confinement is consistent with that found in Australia [21], and contrary to that reported in the United Kingdom [22,23], where prescriptions were observed to increase after the onset of the pandemic. This may be due to differences in the clinical practice guidelines of these countries [24]: in the case of the United Kingdom, the fact that prophylactic treatment against infective endocarditis is not envisaged [25] indicates that lockdown could have increased antibiotic prescriptions in an attempt to delay dental care in non-urgent treatments [26]. However, in countries where prophylaxis with antibiotics is standard practice, the limiting of medical visits during lockdown also limited the prescribing of antibiotics as prophylaxis. 

The increase observed after confinement can be attributed to the delay in treatment arising from confinement (owing to the lack of in-person care and/or patient fear) [27,28] which, due to the inherent progression of untreated processes, led to more severe clinical profiles [29]. This effect was also observed in other diseases (cancers, cardiovascular diseases, diabetes) [30,31,32].

Before the pandemic, a downward trend was observed in antibiotic prescribing by this group of professionals, similar to that reported by studies undertaken elsewhere [33] and the trend at a national level in Spain [34,35]. 

Amoxicillin is the leading antibiotic used by primary-care dentists, a finding in line with the results of other studies [33,35,36,37]. We detected that amoxicillin with clavulanic acid is the second leading antibiotic, ranking almost as high as amoxicillin prescribed alone, even though it is not the antibiotic of choice in most clinical guidelines [38,39,40]. Azithromycin ranks third in the prescribing order, despite being an antibiotic in the Watch group [14] and its use should be limited, due to the greater risk of the appearance of antibiotic resistance. The use of antibiotics such as clindamycin has been associated with an increased risk of *C difficile* infections. Some studies indicate that approximately 15% of community C difficile infections may be caused by odontogenic antibiotic prescribing [41]. Other macrolides, such as azithromycin or clarithromycin have been associated with a higher frequency of adverse reactions [42]. The use of WATCH-type antibiotics as well as other antibiotics, such as clindamycin, should be avoided because of their impact on antimicrobial resistance [14]. These findings are especially interesting, in that they could be the target for interventions that aim to improve dental prescribing. 

We found no study that assessed the impact of the pandemic on antibiotic prescribing by dentists according to the classification proposed by the WHO, nor any study that applied this classification to analysing prescriptions in this field. This prevents us from comparing our results to those of previous studies and would indicate that this classification should also be used in studies on dentists.

### Strengths and Weaknesses of the Study

The principal advantage of this study is the methodology used. The ITS design makes it possible to analyse the impact of the COVID-19 pandemic on antibiotic prescribing [43,44].

By way of limitations, the nature of the study per se should be mentioned. The model used assumes a linear trend in the outcome, and this assumption may only hold over short intervals. Furthermore, the introduction of data broken down by month means that neither individual changes nor the impact of the pandemic on specific days (at the beginning or end of lockdown) can be shown. The study was carried out in one region of Spain; therefore, this geographical limitation could compromise the generalizability. Another study limitation lies in the fact that we do not know whether the initial decrease in the number of prescriptions might have been linked to an increase in the DDD of antibiotics prescribed by dentists working in the private sector or in hospital emergency services. Another study limitation is that the DDD of antibiotics are not correlated to the numbers of in-person consultations and telephone consultations during the pandemic.

Lastly, it should be stressed that the type of data analysed do not record the reason for prescribing, which means that our design does not allow for the appropriateness of prescriptions to be assessed. Thus, there is a need for further studies that would enable long-term changes in trend to be analysed.

The results of this study demonstrate the usefulness of using the WATCH classification to monitor appropriate antibiotic prescribing. These results could be disseminated through specific educational interventions for dentists to raise awareness for the prudent use of antibiotics.

## 4. Materials and Methods

### 4.1. Settings

Galicia is a region situated in the north of Spain, with an area of 29,434 km^2^ and a population of 2.7 million. Healthcare for close to 100% of the population is covered by the Galician Health Service (Servizo Galego de Saúde/SERGAS). Dentistry forms part of primary care, with services including the examination of the oral cavity, treatment of acute processes and dental emergencies, oral surgeries, and preventative treatments. For the remaining dental procedures, such as root canal work or prosthetic rehabilitation, patients have to use a network of private clinics, which also provide the range of treatment offered by SERGAS dentists. This study was conducted using antibiotic prescription data relating to the dentists, some 100 in all, who work at SERGAS primary care buccodental health units [45]. In Spain, medications are dispensed at pharmacies and, in the case of antibiotics, are only available on prescriptions issued by dentists among other health professionals.

### 4.2. Study Design and Data Collection

We used a natural, before-and-after, quasi-experimental design, with monthly data for the period from January 2017 to May 2021. This design, which enables causal effects to be estimated after controlling for baseline levels and trends [46], is suitable for assessing the impact of the pandemic and confinement, since these are time-delimited interventions, and the longitudinal nature of the data provide the analysis with a special robustness [47,48].

The drug class targeted by this paper comprises antibacterials for systemic use (Anatomical Therapeutic Chemical code J01) [49], prescribed in outpatient settings by dentists in the public health sector.

### 4.3. Data Source 

For study purposes, antibiotic prescription data were obtained from the SERGAS Pharmacy Information System for Complex Pharmacy Service Analyses (Sistema Corporativo de Información de Análisis Complejos Prestación Farmacéutica). This shows all medications dispensed by community pharmacies in the health area, based on official prescriptions billed to the Public Health Service. Data from primary-care dentists were selected. There was no change in the system across the study period.

### 4.4. Definition of Variables

We calculated the number of monthly defined daily doses (DDD), defined as the average maintenance dose per day for a drug for its main indication [50].

The World Health Organisation (WHO) suggests the use of the Access, Watch, Reserve (AWaRe) classification issued in 2019 and updated in 2021, for an improved evaluation and monitoring use of antibiotics. This tool makes it possible to reduce antibiotic resistance and access to all medicines [14]. The Watch group includes 110 first- or second-choice antibiotics that display a higher resistance potential when compared with the Access group, and thus should be strictly monitored and restricted to the limited indications, as well as prioritised as major targets for stewardship programs.

### 4.5. Statistical Analysis

An interrupted time series (ITS) analysis model based on a segmented regression approach [44,51,52,53] was purpose-designed to analyse the differences between monthly prescribed DDD of antibiotics [44]. This design allows for effects to be estimated by controlling for baseline levels and trends [46]. A segmented linear regression analysis model [47] was designed to analyse the differences observed from January 2017 to May 2021 in terms of prescribed antibiotics. The independent variables were defined as: time (t:1, 2, 3, 53); a binary variable (COVID) taking values of “0” before June 2020, and “1” after July 2020, corresponding to the time in which the effect of the pandemic was measured; a binary variable (lockdown) taking values of “1” in the months of lockdown (March 2020 to June 2020), and “0” in the other months; a variable for the time elapsed since the first COVID-19 case, which took the value of “0” before; and the values of “1”, “2”, “3”, corresponding to the months from July 2020 to May 2021. To identify possible seasonal changes in antibiotic prescribing, the X-13ARIMA-SEATS procedure was applied [53]. This method is an adaptation of the US Census Bureau X-13–Auto-Regressive Integrated Moving Average (ARIMA) model which produces a seasonally-adjusted time series. The Cumby-Huizinga test was used to check for autocorrelation (10 autocorrelation lags were tested). Based on the results of these tests, lags of the dependent variable were introduced to correct for autocorrelation. The relative changes due to immediate impact and their 95% CI were calculated according to the Boostrap methods described by Zhang [54]. A total of 10,000 replicates were simulated.

All analyses were performed using the free R statistical software environment (version 4.0.5) [55], except for the Cumby–Huizinga test, which was carried out with Stata version 12 [56]. 

## 5. Conclusions

The COVID-19 pandemic led to a decrease in antibiotic prescribing by primary-care dentists during the initial months, followed by an upward trend. This change was likewise observed for Access and Watch antibiotics. Due to the high percentage of antibiotic prescriptions issued by these professionals, it may well be of interest if future studies that analysed antibiotic prescribing among dentists were to use the WHO AWaRe classification.

## Figures and Tables

**Figure 1 antibiotics-11-01018-f001:**
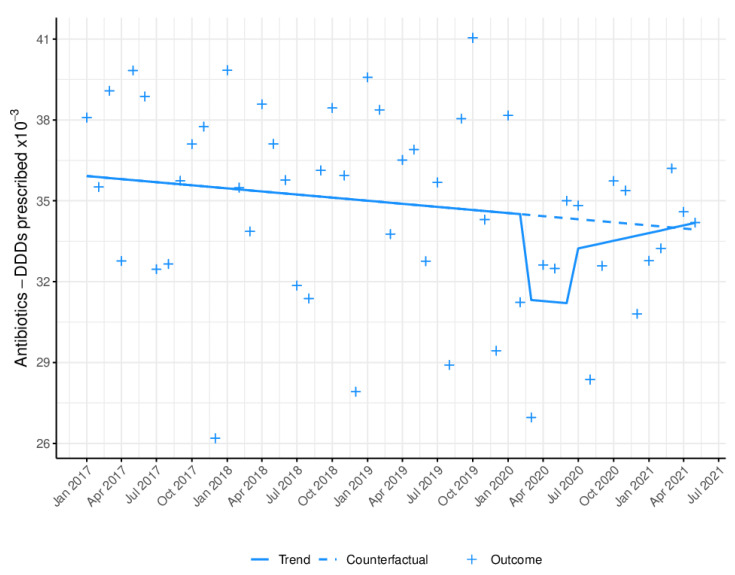
Monthly antibiotic prescribing trend in defined daily doses (DDD). Blue line (−): trend, after model adjustment; blue dashed line: expected trend with no COVID-19 outbreak; +: monthly antibiotic prescribing in DDD.

**Figure 2 antibiotics-11-01018-f002:**
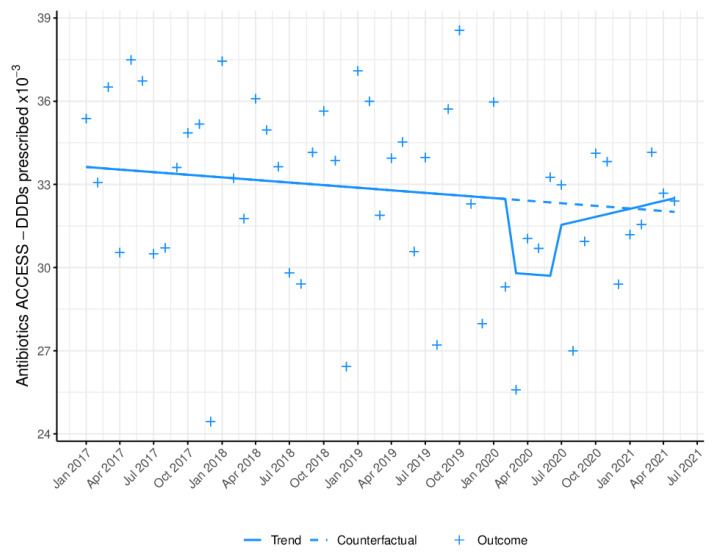
Monthly WHO Access antibiotic prescribing trend in defined daily doses (DDD). Blue line (−): trend after model adjustment; blue dashed line: expected trend with no COVID-19 outbreak; +: monthly antibiotic prescriptions in DDD.

**Figure 3 antibiotics-11-01018-f003:**
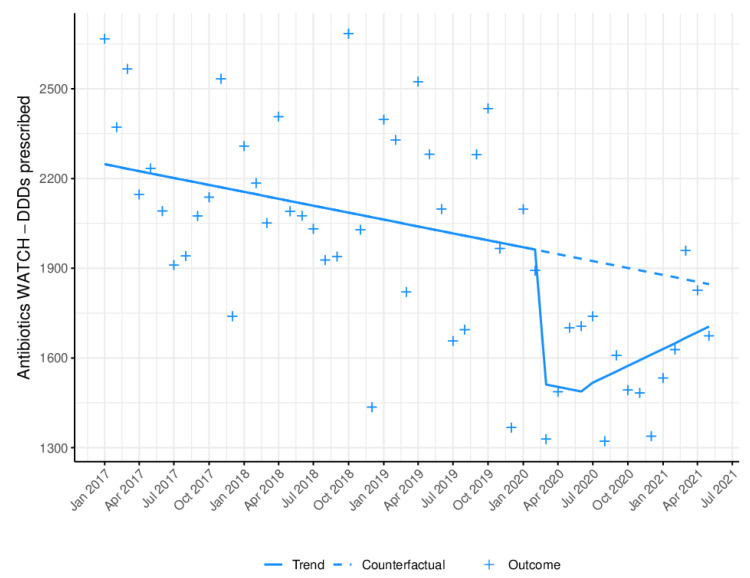
Monthly WHO Watch antibiotic prescribing trend in defined daily doses (DDD). Blue line (−): trend, after model adjustment; blue dashed line: expected trend with no COVID-19 outbreak; +: monthly antibiotic prescriptions in DDD.

**Table 1 antibiotics-11-01018-t001:** Antibiotic consumption before, during and after lockdown [14].

	Pre-Lockdown	Lockdown	Post-Lockdown
	(DDD)	(DDD)	(DDD)
**Total Antibiotics**			
Mean (SD)	3,5344.73 (3606.63)	3,1767.37 (3406.07)	3,3517.72 (2329.37)
Median	35,856.17	32,554.32	34,190.74
PCT25, PCT75	3,2734.07, 3,8218.76	2,8342.59, 3,4405.20	3,2584.43, 3,5381.73
**Access Antibiotics**			
Mean (SD)	3,3170.23 (3360.40)	3,0147.09 (3243.56)	3,1839.90 (2181.16)
Median	33,900.87	30,869.28	32,398.21
PCT25, PCT75	3,0569.86, 3,5783.69	2,6866.07, 3,2705.93	3,0942.64, 3,3819.73
**Watch Antibiotics**			
Mean (SD)	2116.26 (309.18)	1555.75 (182.75)	1600.35 (195.29)
Median	2094.33	1594.00	1609.00
PCT25, PCT75	1936.13, 2340.00	1368.13, 1705.13	1483.00, 1739.17

SD: standard deviation; PCT25: percentile 25; PCT75: percentile 75; DDD: defined daily dose.

**Table 2 antibiotics-11-01018-t002:** Segmented regression analysis of interrupted time series data on antibiotic prescribing [14].

	Antibiotic Prescribing (DDD)		
	Overall	Access	Watch
**Per Month Trend in Pre-Lockdown Period**			
B	−42.39	−35.09	−7.41
95%CI	−96.47; 11.69	−87.35; 17.16	−13.52; −1.30
*p*-value	0.1214	0.183	0.0187
**Effect of Lockdown (1)**			
B	−3572.73	−2976.5	−651.02
95%CI	−6018.90; −1126.57	−5304.34; −648.65	−891.02; −411.01
*p*-value	0.0051	0.0133	<0.0001
**Post-Lockdown Change (1)**			
B	−1363.82	−991.28	−688.91
95%CI	−4001.04; 1273.40	−3537.90; 1555.34	−1029.54; −348.28
*p*-value	0.3033	0.4373	0.0002
**Change Per Month in Trend in Post-Lockdown Period (2)**			
B	142.74	136.05	29.15
95%CI	−193.39; 478.86	−190.25; 462.35	−2.28; 60.58
*p*-value	0.3971	0.4057	0.0682

DDD: defined daily dose; B: non-standardized coefficient; CI: confidence interval. 1. With respect to a pre-lockdown period. 2. With respect to a pre-lockdown trend.

## Data Availability

The datasets used and/or analysed during the current study are available from the corresponding author on reasonable request.

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
