# Peer review of "Impact of the COVID-19 Pandemic on Antibiotic Prescribing by Dentists in Galicia, Spain: A Quasi-Experimental Approach"

_antibiotics, 2022, doi:10.3390/antibiotics11081018_

Round 1

Reviewer 1 Report

Rodriguez-Fernandez et al. reported on antibiotic prescription by dentists during the COVID-19 pandemic. The size of the data set is sufficient. However, the impact of the findings might be limited. Some sentences are incomplete or inconclusive, in particular in the abstract section. The discussion is sound and the use of macrolide antibiotics is concerning, thus being an interesting finding.   MAJOR: 1. 133-135" Hence, the suspension of non-urgent surgical treatments (such 133 as extractions of asymptomatic teeth) may probably have led to a decrease in the prescrib- 134 ing of antibiotics associated with these types of procedures, and of Watch-type antibiotics 135 in particular" - A decrease in antibiotics prescription due to lower numbers of surgical procedures is consistent with other reports, but the consequences of the findings might be limited - The upward trend after the pandemic might be due to increased patient consultations, which have been postponed during the pandemic 2. DDD of antibiotics should be correlated to the numbers of in-person consultations and telephone consultations during the pandemic 3. Table 1: Please include DDD for the most commonly used antibiotics: amoxi, amoxi-clav, clinda   MINOR: 1. Please carefully check for spelling mistakes and incomplete sentences: e.g., "To assess the impact of the COVID-19 pandemic on antibiotic prescrib- 19 ing by dentists, and to review antibiotic consumption according to the WHO Access, Watch, Reserve 20 classification." and "The WHO AWARE classification for antibiotics can 29 be interesting to analyse the prescription in dentists to monitored their appropriate use. "and "Antibiotic resistance is one of the most pressing public health problem worldwide, 35 due to its impact on mortality, morbidity, and healthcare costs"...   2. Please elaborate on the disadvantages of macrolide antibiotics regarding coverage of pathogens and the use in endocarditis prophylaxis 3. Please avoid colloquial phrases, e.g., "needless to say"

Reviewer 2 Report

In the manuscript of Rodríguez-Fernández et al., the principal study's aim was to assess the impact of  the pandemic on antibiotic prescribing by dentists whose practices come within the Spanish National Health System (NHS) in north-west Spain, and the secondary aim was to try and assess the impact of COVID-19 on the prescribing of those antibiotic groups which are most closely related with the development of antibiotic resistance.

The data were important, because the outcomes showed an immediate significant decrease in overall antibiotic prescribing by primary-care dentists during lockdown, followed by a non-significant upward trend for the next year.

Therefore, I have no further comments against the manuscript

Author Response

Thank you very much for your time and for reading our article. We appreciate your valuable comments 

Reviewer 3 Report

The authors report the resulst of a quai experimental study on antibiotic prescribing by dentists according to AWARE classification. The study is well designed and the results well presented.

Few minor comments:

1. Limitations: study geographically restricted which limits generalizability

2. Discussion:

- The authors could add a comment on the utility of their study. Why their results are worth publishing and whether they expect to diffuse them to dentists with the view of rationalising antibiotic prescribing

Round 2

Reviewer 1 Report

Thank you for the revised version. The limitations are now more adequately described. However, since data on the number of consultations is not available, a decline in antibiotic prescription might well be explained by lockdown measures and anxiety of patients leading to a lower number of consultations. Therefore, the impact of this analysis is very limited.